# Large-scale gene losses underlie the genome evolution of parasitic plant *Cuscuta australis*

Guiling Sun[1,2], Yuxing Xu [1,3,4], Hui Liu [1], Ting Sun[2], Jingxiong Zhang [1], Christian Hettenhausen[1], Guojing Shen [1], Jinfeng Qi [1], Yan Qin[1], Jing Li [1], Lei Wang [1], Wei Chang [1], Zhenhua Guo[3], Ian T. Baldwin [5] & Jianqiang Wu [1]

Dodders (*Cuscuta* spp., Convolvulaceae) are root- and leafless parasitic plants. The physiology, ecology, and evolution of these obligate parasites are poorly understood. A high-quality reference genome of *Cuscuta australis* was assembled. Our analyses reveal that *Cuscuta* experienced accelerated molecular evolution, and *Cuscuta* and the convolvulaceous morning glory (*Ipomoea*) shared a common whole-genome triplication event before their divergence. *C. australis* genome harbors 19,671 protein-coding genes, and importantly, 11.7% of the conserved orthologs in autotrophic plants are lost in *C. australis*. Many of these gene loss events likely result from its parasitic lifestyle and the massive changes of its body plan. Moreover, comparison of the gene expression patterns in *Cuscuta* prehaustoria/haustoria and various tissues of closely related autotrophic plants suggests that *Cuscuta* haustorium formation requires mostly genes normally involved in root development. The *C. australis* genome provides important resources for studying the evolution of parasitism, regressive evolution, and evo-devo in plant parasites.

[1] Department of Economic Plants and Biotechnology, Yunnan Key Laboratory for Wild Plant Resources, Chinese Academy of Sciences, Kunming 650201, China. [2] Institute of Plant Stress Biology, State Key Laboratory of Cotton Biology, Department of Biology, Henan University, Kaifeng 475001, China. [3] The Germplasm Bank of Wild Species, Kunming Institute of Botany, Chinese Academy of Sciences, Kunming 650201, China. [4] University of Chinese Academy of Sciences, Beijing 100049, China. [5] Max Planck Institute for Chemical Ecology, Jena 07745, Germany. These authors contributed equally: Guiling Sun, Yuxing Xu, Hui Liu. Correspondence and requests for materials should be addressed to J.W. (email: wujianqiang@mail.kib.ac.cn)

About 1% of the flowering plants are haustorial parasites[1], and some are responsible for severe yield losses in many crops. Although the evolutionary history of plant parasitism remains elusive, all parasitic plants, except the mycoheterotrophs, use a specialized haustorial organ to extract water and nutrients through vascular connections with the hosts[2]. The *Cuscuta* spp. (dodders) are typical obligate shoot parasites widely distributed worldwide comprising ~194 species, and is the only genus of parasites in Convolvulaceae (Solanales). *Cuscuta* spp. exhibit massive changes of their body plans, being leaf- and rootless throughout their lifecycles (Fig. 1a). They contain trace amounts of chlorophyll, but cannot sustain themselves from their own photosynthesis. Although whether *Cuscuta* plants are hemi- or holoparasites remains debatable, they could be considered to be transitioning from hemiparasitism to holoparasitism. Through haustoria, dodders not only obtain water and nutrients, but secondary metabolites, mRNAs, and proteins from their host plants[3–5]. These features make *Cuscuta* an important model to elucidate plant–parasite interactions and the evolution of plant parasites.

Here we sequence the genome of *Cuscuta australis*. Our analyses reveal that the genome of *C. australis* experienced massive gene losses, including important genes involved in leaf and root development, flowering-time control, as well as defense against pathogens and insects. Comparison of *Cuscuta* haustorium/prehaustorium gene expression patterns with the tissues from closely related autotrophic plants suggests that *Cuscuta* haustorium formation likely requires genes that are normally involved in root development.

## Results

**Genome assembly.** The genome size of *C. australis* was estimated to be 272.57 Mb from *k*-mer analysis. We next generated 26.6 Gb (97.6-fold coverage) of *C. australis* genome sequences from a single-molecule real-time (SMRT) sequencing platform. The *C. australis* genome sequence includes 249 contigs (N50 = 3.63 Mb), and these contigs were further assembled to form 103 scaffolds (N50 = 5.95 Mb; Supplementary Table 1). In total, 264.83 Mb (219 contigs) of nuclear sequences (97.16% of the estimated genome size) and 1.9 Mb (30 contigs) of organellar sequences were acquired (Supplementary Table 2). The accuracy and heterozygosity were estimated to be 99.99% and 0.013%, respectively (Supplementary Table 3). A total of 155 Mb of repetitive elements were identified in the *C. australis* nuclear genome (Supplementary Table 4). Comparison with *Ipomoea nil* (Japanese morning glory; also Convolvulaceae), which is relatively closely related, indicated similar proportions of different types of repetitive elements between the two genomes (Supplementary Table 4), although the *I. nil* genome contains many more repeats (Supplementary Table 4), consistent with its larger genome size (734 Mb). LTR (long terminal repeats) retrotransposons are the most dominant type of repeats in both *C. australis* and *I. nil*

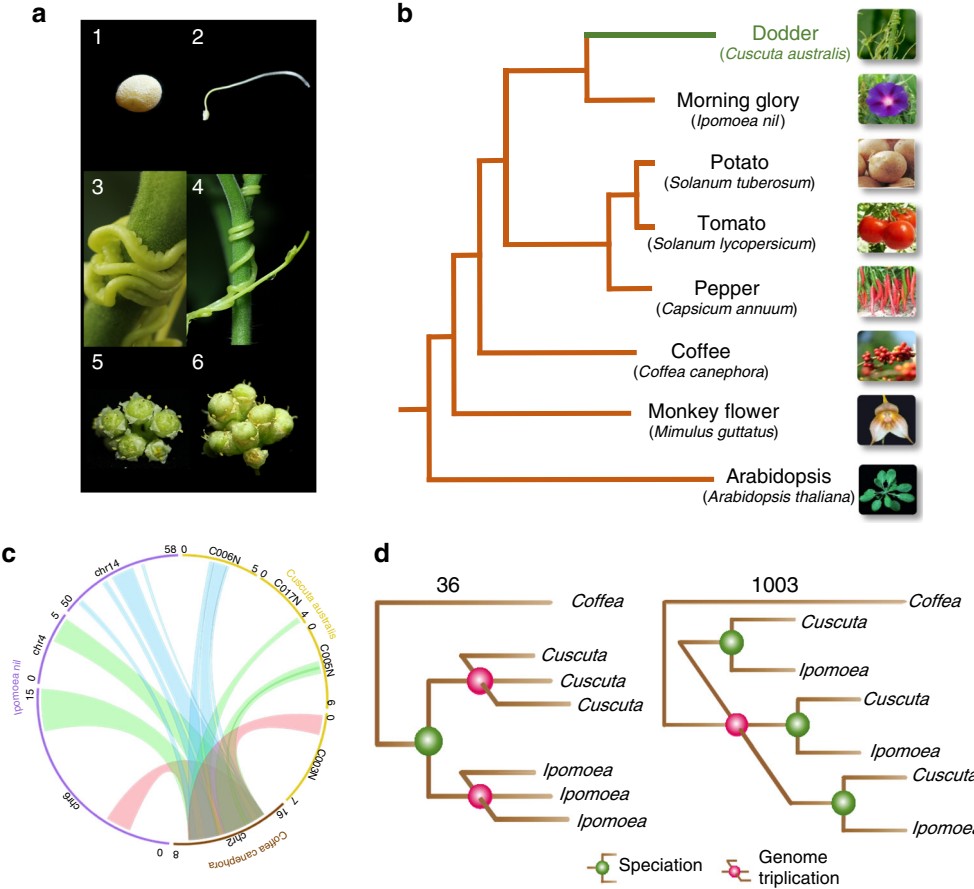

**Fig. 1** Morphological traits and genome structure of *C. australis*. **a** Photographs of *C. australis* seed (1), seedling (2), vines twining around the wild tomato *Solanum pennellii* (3 & 4; partial haustoria can be seen in 3), flowers (5), and seed capsules (6). **b** Phylogenetic tree generated from genome-wide one-to-one orthogroups (bootstrap values for all clades are 100%). **c** Circos plot of a set of syntenic genome segments of *C. australis*, Japanese morning glory, and coffee. Numbers besides terminals of each karyotype denote the start and end of chromosome segment or contigs with unit of Mb. **d** Numbers of gene clades (shown on top of the trees) supporting different hypotheses on the order of speciation and whole-genome triplication event in the *Cuscuta* and *Ipomoea* lineage

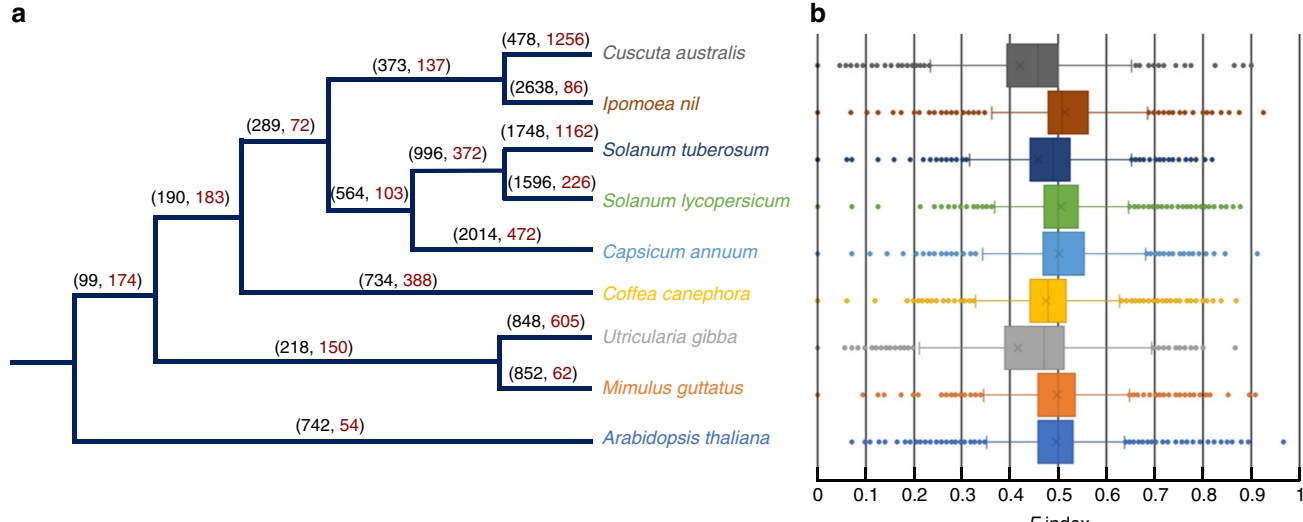

**Fig. 2** Expansion and contraction in *C. australis* gene families. **a** Significantly expanded and contracted gene families. Brackets above each branch indicate numbers of expanded (in black, before comma) and contracted (in red, after comma) gene families. **b** Tukey boxplot overview of the differences among the gene numbers of the conserved gene families, based on the *F*-index values (*F*-indices range from 0 to 1; when *F* = , < , or > 0.5, the gene number in the given gene family is equal to, smaller, or greater than the average size of this gene family in all species). The left and right sides of the boxes are the first and third quartiles, respectively; means and medians of the data are shown as an "×" and the bands in the boxes, respectively; for each box, the whiskers represent the smallest and biggest datum that are still within 1.5 times interquartile range of the lower and upper quartile, and the outliers are shown as dots

(Supplementary Table 4), and we specifically inspected the *LTR/Gypsy* and *LTR/Copia* superfamilies. These two genomes exhibit large differences in *LTR/Gypsy* (84,083 in *I. nil* and 24,453 in *C. australis*) and *LTR/Copia* (96,355 in *I. nil* and 43,853 in *C. australis*) copy numbers; moreover, the sequences of *LTR/Gypsy* and *LTR/Copia* members have also diverged as revealed by phylogenetic analysis (Supplementary Fig. 1).

Based on de novo gene structure prediction, homology comparison, and transcript data of both *C. australis* and the closely related relative *C. pentagona*[6], 19671 genes could be annotated.

**Phylogeny analysis.** The phylogenetic position of *Cuscuta* was determined using 1796 one-to-one orthogroups (Supplementary Data 1b) identified from *C. australis*, *Arabidopsis thaliana*, and six lamiids plants—the Solanales *Ipomoea nil*, *Solanum tuberosum* (potato), *Solanum lycopersicum* (tomato), and *Capsicum annuum* (pepper), the Gentianales *Coffea canephora* (coffee), and the Lamiales *Mimulus guttatus* (monkey flower) (for simplicity, these seven autotrophic species are collectively named 7Ref-Species). Consistent with their phylogenetic relationship, *Cuscuta* forms a sister group with *Ipomoea*, and we estimated that these two lineages split ~33 million years ago (Supplementary Fig. 2). Notably, *Cuscuta* shows a much longer branch than does *Ipomoea* (Fig. 1b), providing genome-wide evidence consistent with the hypothesis that parasitic plants evolve rapidly[7, 8]. This result is statistically significant by both two cluster analysis and relative rate test (Supplementary Table 5 and Supplementary Table 6). Previously, a whole-genome duplication (WGD) event was detected in *Ipomoea*[9]. The syntenic blocks and trees of the syntenic gene groups of *Cuscuta* and *Ipomoea* vs. *Coffea* genome indicate that *Cuscuta* and *Ipomoea* experienced a whole-genome triplication event before their divergence from a common ancestor (Fig. 1c, d, Supplementary Fig. 3 to 5, Supplementary Table 7).

**Contractions and expansions of gene families.** Parasitism and large changes of body plan in *Cuscuta* suggest that many gene

families might have experienced substantial alterations in sizes, including those that function in leaf and root physiology. To study gene family expansion and contraction, a rigorous bioinformatic pipeline was adopted to identify gene families (details see Supplementary Note 5). In addition, the genome of *Utricularia gibba* (Lentibulariaceae; an aquatic carnivorous bladderwort plant) was included in the analysis, given that *U. gibba* also exhibits large changes in body plan (e.g., no true roots). We identified a total of 13981 gene families in *C. australis*, *U. gibba*, and the 7Ref-Species (Supplementary Data 1a); among these, 1256 and 478 families in *C. australis* and 605 and 848 families in *U. gibba* were found to have had significant contractions and expansions, respectively, revealed by a maximum-likelihood analysis (Fig. 2a; Supplementary Data 1a). Moreover, box plots of the *F*-indices (details see Supplementary Note 5), which describe the differences among the gene numbers of the conserved gene families in the 7Ref-Species (namely, in *Arabidopsis* and at least five of the six remaining autotrophic species), indicate that in *C. australis* and *U. gibba*, gene numbers in 72% and 62% of the conserved gene families are below the averages, respectively (Fig. 2b).

**Overall gene losses.** The drastic contractions of gene families in *C. australis* and *U. gibba* suggest considerable gene losses in their genomes. Next, BUSCO analysis[10] was carried out to map the 1440 conserved orthologs in land plants to the genomes of 7Ref-Species, *C. australis*, and *U. gibba*. Consistent with their contracted gene families, the missing BUSCOs in *C. australis* and *U. gibba* (16.30% and 13.70%, respectively) are more than those in the 7Ref-Species (1.40% to 8.50%) (Supplementary Table 8).

To identify specific orthologous genes that are lost in *C. australis* and *U. gibba*, we developed a stringent genome-wide analysis pipeline to divide each gene family into small orthogroups using a method combining phylogenetic and syntenic analysis (details see Supplementary Note 6) and the functional annotations were assigned using *Arabidopsis* genome as the reference (Supplementary Fig. 6). This analysis resulted in 21487 orthogroups (Supplementary Data 1b). Among the 11995

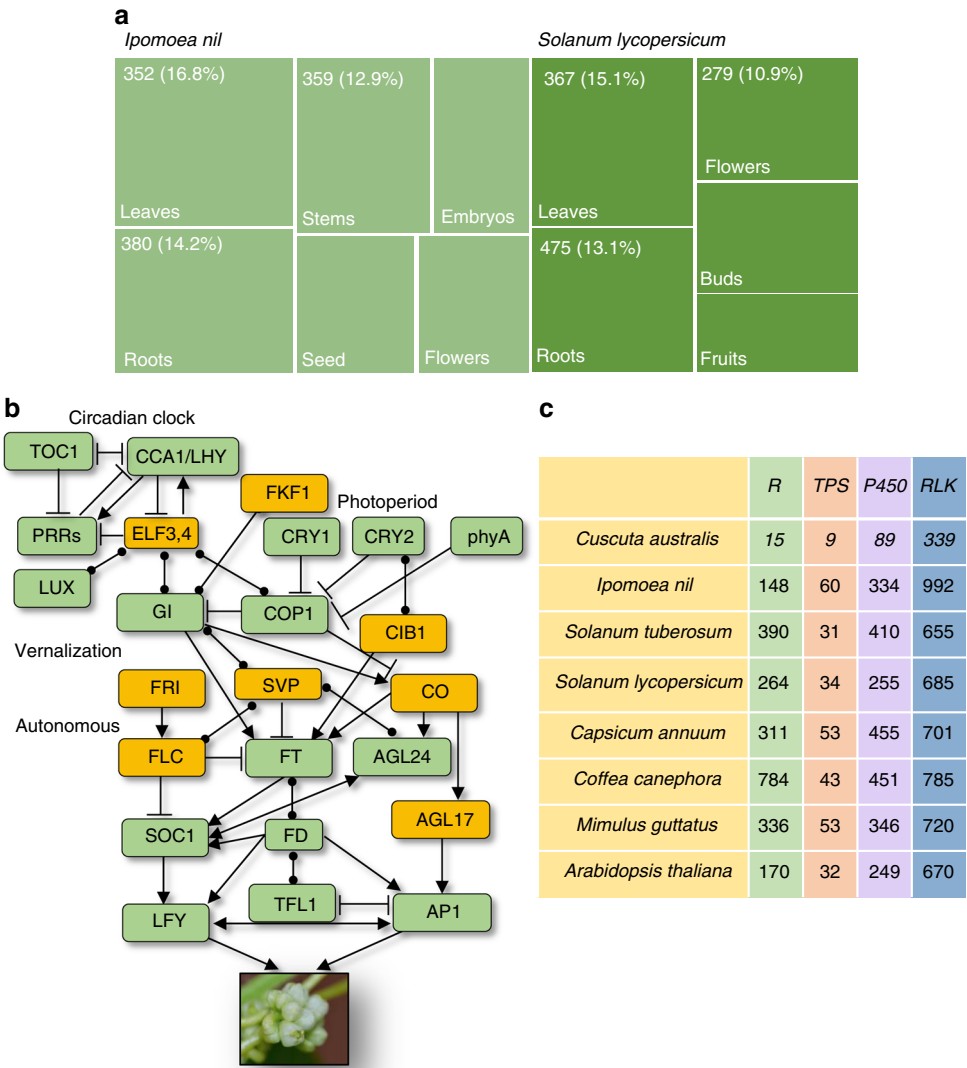

**Fig. 3** Gene losses in *C. australis*. **a** The principally expressed tissues (PETs) of the respective orthologs of *C. australis* lost genes in *S. lycopersicum* and *I. nil*. The boxes represent different tissues. The respective PETs of the orthogroups, which have no *C. australis* members, were identified in *S. lycopersicum* and *I. nil*. The numbers of orthogroups, which have PETs in leaves, roots, flowers, and other tissues, are shown in the boxes, and the respective percentages (proportional to the areas of the boxes) indicate the ratios between these indicated numbers and the numbers of all orthogroups whose PETs were identified to be the corresponding tissues of *I. nil* and *S. lycopersicum*. **b** Simplified gene network controlling flowering time. Genes in green boxes are retained in *C. australis*, and the lost ones are in orange boxes. Arrows and T-ends represent promoting and inhibiting genetic interactions, respectively, and round dots at both ends symbolize genetic interactions with unknown directions. **c** Numbers of genes in the gene families of *R* genes, *TPS*s, *P450*s, and *RLK*s

conserved orthogroups in the 7Ref-Species, there are 1402 and 1555 orthogroups whose *C. australis* and *U. gibba* members are absent, respectively (Supplementary Data 1b and 1c). Strikingly, 563 orthogroups have no *C. australis* and *U. gibba* orthologs, whose functions include phytohormone pathways, nutrient uptake, defense response, and root hair development (Supplementary Data 2a); 839 orthogroups specifically lost their members in *C. australis*, and these genes are mainly involved in response to light, photosynthesis, chloroplast RNA processing, and adventitious root development (Supplementary Data 2b); 992 orthogroups specifically have no members of *U. gibba*, and genes in these orthogroups are involved in signaling, response to stimuli, and protein modifications, among others (Supplementary Data 2c). In addition to the lost genes, in *C. australis* genome we identified 1168 pseudogenes, which contain frame-shifts or premature stop codons. Among these, we found five flowering-

time-related genes and 13 photosynthesis-related genes (Supplementary Data 3a).

We next inspected the tissue-specific expression patterns of the *S. lycopersicum* and *I. nil* (the closest autotrophs to *Cuscuta* among the 7Ref-Species) orthologs in the 1402 orthogroups, whose *Cuscuta* members are lost (Supplementary Data 1d). It was found that these orthologs' principally expressed tissues (the tissues, in which the expression levels of a given gene are at least 1-fold greater than the averages of its expression levels in other tissues, are defined as the principally expressed tissues for that gene) are the leaves and root of both *S. lycopersicum* and *I. nil* (Fig. 3a). These data are consistent with the leaf- and rootless body plan of *Cuscuta*.

**Loss of genes for leaf and root development**. Next, the *C. australis* genome was specifically searched for genes that mediate leaf

and root development, and it was found that a number of important genes involved in leaf and/or root development are absent (Supplementary Data 1c): The orthologs of (1) *LCR* (leaf shape and vein formation); (2) *TRN1/LOP1* (leaf patterning and lateral root development); (3) *PLT1*, *2*, and *5* (specification of root stem cell niche); (4) *SMB* (lateral root cap maturation); (5) all the *CASP* genes which are required for Casparian strip formation; (6) *WAK1* to *5* (root cell expansion); (7) *WOX3*, *5*, and *7* (embryonic patterning, stem cell maintenance, and organ formation).

**Loss of genes for nutrient uptake.** We found that many genes involved in potassium (K), phosphate (P), and nitrate (N) uptake from soil are lost in *C. australis*. For $K^+$ uptake: the orthologs of *HAK5* (a high affinity $K^+$ transporter), *KAT3/AtKC1* (a general regulatory negatively modulating many inward Shaker $K^+$ channels), and *CHX20* (a $Na^+(K^+)/H^+$ antiporter) are absent in *C. australis*. The following orthologs important for P uptake are also lost: *PHO2* (P uptake and translocation), *PHT2;1* (a chloroplast P transporter), and *SPX3* (P signaling). Several genes involved in N uptake are missing as well, such as the orthologs of *NAXT1* (a nitrate efflux transporter), and *NLP7* (a positive regulator for nitrate-induced gene expression); moreover, while the low-affinity nitrate transporters *NRT1.1* and *1.2* are retained, the high-affinity nitrate transporters *NRT2.1* and *2.2* and *NRT3.1* are absent.

**Loss of photosynthesis genes.** *C. australis* has very limited photosynthetic capability (Supplementary Fig. 7). Among 248 photosynthesis-related genes in *Arabidopsis* annotated by GO, 38 *C. australis* orthologs are missing (Supplementary Data 1c). The plastome of *C. australis* (Supplementary Fig. 8) appears to have gene contractions (81 genes remain, while the plastome of *I. nil* harbors 104 genes), and is similar to those of the previously sequenced *Cuscuta* species[11, 12] (Supplementary Data 4), especially *C. obtusiflora* and *C. gronovii*, all of these belong to the subgenus *Grammica* with *C. australis*. Notably, all these *Cuscuta* plastomes lack *ndh* genes encoding proteins for forming the NADH dehydrogenase complex functioning in electron cycling around photosystem I under stress. These data concur with the generally accepted notion that *ndh* genes are first to be lost in the initial stage in the evolution of parasitism, apparently due to relaxed selective constraint[13] (Supplementary Data 4).

**Loss of genes controlling flowering time.** Leaves play a critical role in perceiving environmental signals and thereby modulate the physiology of their own as well as other plant parts, such as activating flowering[14]. We speculate that the leafless dodders may have unique means of regulating flowering time and many flowering-time-control genes may have been lost. A database of flowering-time gene networks in *Arabidopsis* was recently constructed[15]. We found that among the 295 coding genes listed in this database, 26 are lost (Supplementary Data 1c), including the well-known *FLC*, *FRI*, *SVP*, *AGL17*, and *CO* (Fig. 3b). Moreover, the circadian clock genes *ELF3* and *4*, *ARR3* and *4*, and *CDF1* and *3* are also missing. *FKF1* and *CIB1*, which are essential in the photoperiod pathway, are also lost (Fig. 3b). Flowering time is controlled by multiple pathways[16], and it appears that the vernalization, temperature, autonomous, circadian clock, and photoperiod pathway all seem to be nonfunctional (Fig. 3b). The regulation of *C. australis* flowering time is particularly interesting to explore further.

**Loss of defense-related genes.** Leaves and roots are the most common sites of pathogen and insect infestation. We speculate that after the ancestor of *Cuscuta* became leaf- and rootless,

reduced exposure to pathogens and insects relaxed selective constraints on defense-related genes, resulting in their eventual loss. Specifically, four gene families were inspected in detail (Fig. 3c). R (resistance) genes are the critical components of plant immunity. Even though there is a possibility that the ancestor of Convolvulaceae experienced a reduction of *R* genes after the split between Convolvulaceae and other Solanales lineages, as *I. nil* has the smallest number of *R* genes (148; Supplementary Data 3b) among the 7Ref-Species, the reduction of *R* genes in *Cuscuta* is still drastic: *C. australis* genome harbors only 15 *R* genes (Fig. 3c and Supplementary Data 3b). Terpenes function as defenses against insects and pathogens. We predicted 9 terpene synthase genes (*TPSs*) in *C. australis*, and there are 31 to 60 *TPSs* in the 7Ref-Species (Supplementary Data 3c). Many P450 enzymes are involved in the biosynthesis of plant secondary metabolites, which are required for adaptation to biotic and abiotic stresses[17]. While more than 240 P450s are encoded by the genomes of the 7Ref-Species, *C. australis* genome harbors only 89 (Supplementary Data 3d). Receptor-like kinases (RLKs) are important in plant development and resistance to abiotic and biotic stresses. *C. australis* genome comprises only 339 *RLK* genes, much less than in the 7Ref-Species (at least 655 in potato) (Supplementary Data 3e). Consistently, several well-studied genes in plant resistance to pathogens are also absent in *C. australis*, including *EDS1*, *EDS5*, *FMO1*, *SAG101*, and *PAD4*, which are essential for plant resistance to diseases, and *ALD1*, which is critical for systemic acquired resistance.

**Origin of haustorium.** The haustorium is a parasitic plant-specific organ, playing a critical role in establishing parasitism. *C. australis* and *C. pentagona* RNA-seq data[6] were assembled using the *C. australis* genome as the reference (mapping ratios 96% and 75%, respectively; Supplementary Table 9). In *Cuscuta* prehaustoria and haustoria, we identified 2466 principally expressed genes (PEGs; at least one-fold greater than the average of the expression levels in all the other tissues) belonging to 1299 orthogroups (Supplementary Data 1d, Supplementary Data 3f). GO analysis on these PEGs indicated enrichment of biological processes of metabolic process, transport, lignin and xyloglucan metabolism, and transcriptional regulation genes (Supplementary Data 2d). This is consistent with the haustorial function of transporting host substances and the findings that dynamic cell wall remodeling in parasite haustoria is important in the establishment of parasitism[18–20]. The biggest proportion of these 1299 corresponding orthologs' principally expressed tissues in *S. lycopersicum* and *I. nil* were found to be the roots (Supplementary Fig. 9, Supplementary Data 1d). These data imply that the evolution of *Cuscuta* haustorium may be related to expression changes of genes involved in root development. Similarly, comparative transcriptome analyses on three root parasites, *Triphysaria versicolor*, *Striga hermonthica*, and *Phelipanche aegyptiaca*, revealed that parasitism genes are derived primarily from root and floral tissues[18].

Next, we performed a HYPHY analysis to obtain the genes that underwent positive selection and relaxed purifying selection after the divergence between the ancestors of *Cuscuta* and *Ipomoea* lineage, as these genes might be associated with the speciation of *Cuscuta* and evolution of parasitism. GO terms including "response to hormones", "DNA methylation", "regulation of transcription", and "cell wall-related metabolism" were enriched from the 1124 positively selected genes (Supplementary Data 3g, Supplementary Data 2e), and among these, 115 are principally expressed in prehaustoria/haustoria (Supplementary Data 3f), including a pectin esterase, receptor-like kinases, transcription factors, a serine carboxypeptidase, and transporters. We also

found that 3890 genes (Supplementary Data 3h) exhibited signatures of relaxed purifying selection. The enriched GO terms include "terpenoid biosynthetic process", "nitrate assimilation", "photosystem II assembly", and "regulation of signal transduction" (Supplementary Data 2f), and 504 genes with relaxed purifying selection (Supplementary Data 3f) are principally expressed in prehaustoria/haustoria, such as genes encoding subtilisin-like proteases and an ABC transporter.

Gene family expansion, mainly caused by gene duplication, often leads to neofunctionalization among the gene family members and is thought to be an important driving force in the acquisition of novel phenotypes. Thus, we performed GO analysis on the 3099 expanded gene family members of *C. australis* (Supplementary Data 3i, Supplementary Data 2g). It was found that "response to auxin" and "DNA methylation" were enriched from the members of expanded gene families; among them, 109 genes are principally expressed in prehaustoria/ haustoria (Supplementary Data 3f). These data are consistent with the finding that many haustorial genes in Orobanchaceae parasites also experienced relaxed purifying and/or positive selection and may play an important role in the evolution of the parasitic lifestyle[18].

Five positively selected genes from the expanded gene families were found to be principally expressed in haustoria (Supplementary Fig. 10. and Supplementary Data 3f). Among these, one encodes a putative α/ß-hydrolase highly similar to *Nicotiana sylvestris* DAD2/DWARF14 (84% identity), which is the strigolactone receptor in autotrophic plants, implying that neofunctionalization of α/ß-hydrolase genes might also be involved in the parasitization process in *Cuscuta*, as they are in the root parasites *Striga* and *Orobanche*[21].

## Discussion

How plants evolved parasitism is still unclear. Our transcriptomic data and molecular evolution analysis suggest convergent evolution in *Cuscuta* and Orobanchaceae root parasites[18]. The haustorium of *Cuscuta* probably evolved from root tissues and that of Orobanchaceae parasites recruited genes normally involved in the development of root and floral tissues. Moreover, a relatively large fraction of the genes that experienced positive selection and/or relaxed selection are principally expressed in prehausotria/haustoria in both *Cuscuta* and Orobanchaceae, and these genes may be related to parasitization and/or evolution of parasitism.

Our comparative genomic analyses indicate that the *C. australis* genome experienced remarkably high levels of contraction, and this is consistent with *Cuscuta*'s parasitic lifestyle and large changes in body plan—leaf- and rootless as well as intensive dependence on host-derived metabolites and signals. Given the importance of new genes that bring about novel phenotypes, it is possible that the autotrophic ancestor evolved the haustorium from root tissues through the neofunctionalization of duplicated genes and transcriptional reprogramming in the *Cuscuta* lineage. *Cuscuta* is transitioning from hemiparasitism to holoparasitism. Among the recognized major parasite lineages, three contain only hemiparasites, while eight are solely holoparasites[1], implying that holoparasitism may have additional selective advantages over hemiparasitism. Hypothetically, the dramatic changes in body plans, including the degeneration of leaf and root could allow these parasites to reallocate carbon and nitrogen resources, which are required for their development and growth, to reproduction and could also enable the holoparasites to better adjust their physiology according to that of the hosts by eavesdropping on host signaling molecules.

Wicke et al.[13] analyzed the plastomes of nonparasitic, hemi-parasitic, and nonphotosynthetic parasitic plants in Orobanchaceae, and found that the transition from autotrophic plants to obligate parasites relaxes functional constraints on plastid genes in a stepwise manner. Similar analyses could not be done in *Cuscuta*, as it is the only parasitic lineage in the Convolvulaceae. The large body plan changes that were associated with the parasitic lifestyle relaxed selective constraints on several core pathways, such as photosynthesis[13], flowering-time control, root and leaf development, nutrient acquirement, and defense against pathogens and insects. Under relaxed selection pressure, some genes of these pathways were pseudogenized and melted into the background of surrounding DNA because of accumulation of recurrent mutations, or were even deleted from the genome, finally leading to gene losses. In *C. australis* genome, 1168 genes were identified as pseudogenes, although some may still be functional or even have gained new functions[22]. It is likely that the majority of these pseudogenes have degenerated or are in the process of being deleted, following the fate of the lost genes. Indeed, *C. australis* genes that have undergone relaxed purifying selection include those related to terpenoid biosynthetic process (defense), nitrate assimilation (nutrient acquirement), and photosystem II assembly (photosynthesis). Resequencing of other *Cuscuta* species may shed light on the recent gene loss events and the underlying mechanisms (e.g., transposon insertion, fragment deletion, and rapid accumulation of mutations) in *Cuscuta*.

Large scale gene loss is also evident from the genomic data of human parasites *Pediculus humanus humanus* (body louse)[23], *Trichinella spiralis* (a zoonotic nematode)[24], and *Giardia lamblia* (an intestinal protist)[25], and from an *Arabidopsis* pathogen, the obligate biotrophic oomycete *Hyaloperonospora arabidopsidis*[26]. Here, we also detected that *U. gibba* also experienced a large number of gene loss events during its evolution, likely resulting from its body plan changes (mainly loss of root) and lifestyle alteration (carnivory). It would also be of importance to compare the genomes of *Cuscuta* and those of Orobanchaceae root parasites. We expect that Orobanchaceae parasites, especially holoparasites, such as *Orobanche* species, may also have a large number of lost nuclear genes, some of which may bear the same functions as those in *Cuscuta*, such as the genes that function in leaf development and photosynthesis. The *C. australis* genomic data strongly support the notion that regressive evolution, which is associated with extensive gene loss, is likely to be pervasive and adaptive during the evolution of holo- /obligate parasites[27, 28]. Comparative genomics between the genome data of *C. australis* and autotrophic plants provides an important resource for studying genome reduction, regressive evolution, parasitism, and evo-devo in plant parasites.

## Methods

**DNA sample preparation and sequencing**. The seeds of *C. australis* were originally purchased from a Chinese traditional medicinal herbs store in Kunming, China, in 2011, and had been cultivated for five generations in a glasshouse at the Kunming Institute of Botany, Chinese Academy of Sciences. Voucher specimens of *C. australis* can be accessed at the Herbarium of the Kunming Institute of Botany, Chinese Academy of Sciences (accessions Nos. WJQ-001-1 and WJQ-001-3). Seedlings were prepared from seeds of the fifth generation and were infested on soybean plants (for the germination procedure, see Li et al.[29]). DNA isolated from young vines collected from one individual *C. australis* was used for genomic library construction. Three genomic DNA libraries with 350-bp, 2-kb, or 5-kb insertions were constructed for Illumina sequencing. For PacBio sequencing, five DNA libraries with 20-kb insertions were sequenced on a PacBio RS II instrument using the P6v2 polymerase binding and C4 chemistry kits (P6-C4). A total of 26.6 Gb from 1,953,966 reads were obtained by processing 24 single-molecule real-time (SMRT) cells. The average and N50 of SMRT subread length were 9.6 kb and 13.6 kb, respectively.

**Transcriptome sample preparation and sequencing**. *C. australis* tissues of seeds, just-germinated seeds (1 day after imbing), seedlings (5 days after imbing), prehaustoria, haustoria, stems far from haustoria, stems near haustoria, flower buds, flowers, and seed capsules were collected and RNA samples were extracted

from these tissues using the TRI Reagent (Sigma). Libraries were constructed for each tissue according to the TrueSeq® RNA Sample Preparation protocol, and sequenced on an Illumina HiSeq-2500. Sequences are deposited in NCBI under BioProject PRJNA394036.

*C. pentagona* transcriptomic short reads dataset from Ranjan et al.[6] were retrieved from the NCBI Short Read Archive under accession numbers SRR965929, SRR965963, SRR966236, SRR966405, SRR966412, SRR966513, SRR966542, SRR966549, SRR966619 to SRR966622, SRR967154, SRR967164, SRR967181 to SRR967190, SRR967275 to SRR967289, and SRR967291.

**Genome survey**. We used 23.29 Gb of HiSeq reads to estimate the genome features using the GCE[30] (v1.0.0) software based on $k$-mer depth-frequency distribution. A total of 11,700,637,064 17-mers were counted. Given the unique $k$-mer depth of 42, we calculated that the genome size = KmerCount/Depth = 272.57 Mb. The repeat content was estimated to be 58.94% based on $k$-mer depth distribution.

**De novo assembly**. SMRT subreads were corrected, trimmed, and assembled using CANU[31] (v1.3), a genome assembler built on the basis of Celera assembler. Briefly, SMRT subreads were firstly self-corrected based on an overlap-layout algorithm. Erroneous regions of error-corrected SMRT subreads were then trimmed to increase accuracy. We constructed an initial assembly using CANU with those trimmed error-corrected SMRT subreads following the parameters "genomeSize = 273 m errorRate = 0.025" and then used Quiver[32] (v4.0) to generate consensus sequences by aligning SMRT subreads to correct the errors in the assembly. Lastly, we used Pilon[33] (v1.18) to perform the second round of error correction with HiSeq reads from the 350-bp-insert library. The resulted error-corrected assembly was named version 1.0 and used in the subsequent analyses. A hierarchical method was also applied to concatenate adjacent contigs: SSPACE[34] (v 3.0) was first used to build scaffolds using HiSeq data from the mate-pair libraries and the contigs built from the PacoBio data. N50 of the resulted scaffolds reached 5.9 Mb. SSPACE-LongRead[35] (v1-1) was further applied to build superscaffolds with PacBio long reads, nevertheless, no linking information between scaffolds were found and no improvements were acquired (Supplementary Table 1).

**Assembly assessment**. The accuracy and heterozygosity rate of the genome assembly were estimated using the following procedure: adaptors were removed from the short paired-end reads obtained from the 350-bp insert size library, and then the reads were aligned to the PacBio assembly, which had been corrected with Pilon[33] (v1.18), using bowtie2[36] (v 2.2.4) to generate a bam file. A total of 97.9% reads could be mapped. Samtools[37] (v 1.3.1) were used to sort the bam file, and Freebayes[38] (v1.1.0) was applied to call variants. Compared with PacBio assembly as the reference genome, homozygous SNPs or indels were considered to be assembly errors, and heterozygous SNPs or indels were regarded as heterozygous sites. The accuracy was estimated to be 99.99% and heterozygosity was estimated to be 0.013% (Supplementary Table 3). Error rate of the genome assembly before Pilon correction was also estimated using the same set of methods. A total of 980 SNPs and 101825 Indels were corrected after polishing.

**Data availability**. The genome assembly, gene models, and sequence reads are available at the NCBI under the BioProject PRJNA394036, and the former two can also be accessed at http://www.dodderbase.org. Phylogenetic analysis data for detection of gene losses, gene alignments used for selection analysis, and pseudo-gene annotations can be accessed at https://doi.org/10.6084/m9.figshare.6072131. All data are also available from the corresponding author upon request.

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

## Acknowledgements

This work was supported by the Strategic Priority Research Program of the Chinese Academy of Sciences (CAS) (No. XDB11050200; J.W.). The authors also thank Drs. Lei Wang (Institute of Botany, CAS), Hongtao Liu (Institute of Plant Physiology and Ecology, CAS), Jinyong Hu and Chengjun Zhang (Kunming Institute of Botany, CAS), and Zhaojun Ding (Shangdong University) for valuable comments.

## Author contributions

J.W., G.S., and Y.X. designed and conceived research; T.S., C.H., G.S. prepared plant tissues; W.C. performed photosynthesis experiments; T.S., and Y.Q. prepared DNA samples; J.L. and J.Q. prepared RNA samples; Y.X., and H.L. performed genome assembly and genome annotation; Y.X., H.L, and T.S. analyzed data; J.W., G.S., Y.X., H.L., J.Z., L.W., I.T.B., and Z.G. wrote the paper. All authors read and approved the final manuscript.

## Additional information

**Competing interests:** The authors declare no competing interests.

