## [Peer Review File · Nature Communications]

Reviewers' comments:

Reviewer #1 (Remarks to the Author):

Sun et al. report a high quality reference genome for the parasitic plant *Cuscuta australis*. Based on detailed comparative genomics analysis *C. australis* has experienced extreme gene loss presumably related to its parasitic lifestyle, including many genes related to photosynthesis, leaf, flower and root development among others. Many gene families are contracted including those containing R genes and P450 enzymes reflecting relaxed disease pressures. The manuscript is well written and exciting with major insights into genome evolution and parasitic plant biology. I have a few minor comments.

1. What is the nature of missing orthologous genes? If these genes have root or leaf specific expression, it's conceivable they would be absent from the trinity based transcriptome assembly and may be discarded by Pasa during annotation. The loss of flowering and circadian clock genes is particularly fascinating but also unusual, as these plants produce (relatively) normal flowers. The authors identified ~1,283 putative pseudogenes but it is unclear if these 'core' pathway genes have become pseudogenes or have been deleted altogether.
2. The authors assembled a complete 116kb chloroplast genome as well as several other contigs. Other *Cuscuta* lineages have experienced wide scale gene loss (See Braukmann et al, J Exp Bot. 64(4); 977-989) and it would be interesting to report organelle gene loss in *C. australis*.
3. What program was used to align the Illumina data to the draft PacBio contigs prior to Pilon correction? How many indels/SNPs did Pilon correct? The estimated error rate is somewhat circular if Pilon was run based on the Illumina data aligned using BWA and GATK.

Minor

List the versions for all bioinformatics software (Canu, Quiver, Pilon, etc).

Lines 582-588. This is difficult to follow

Line 666 drew to drawn

Robert VanBuren

Reviewer #2 (Remarks to the Author):

The manuscript by Sun et al. describes the genome of the parasitic plant *Cuscuta australis*, and conducts a set of basic comparative genomic analyses. I realize that this work presents an important resource for the field of plant genomics. However, I have several reservations about it to note, some of which of major concern to me. Several claims appear to be based on weak grounds. The authors claim that many of the genomic changes are due to parasitism, precisely to the loss of autotrophy and the "regressive evolution" associated with it. However, except for the ca. 60 photosynthesis gene orthologs that are indeed directly associated with autotrophy, many other changes such as substantial gene family expansions or reductions are not solely attributed to the autotrophic lifestyle but to an extensive change of a plant's body plan. In consequence, the authors oversimplify their data and resulting statements/conclusions. Whether or not the evolution of these gene family truly associate with the lifestyle thus are in the open and it has not been formally tested, in part due to the authors' study design. Disentangling lifestyle effects is everything but trivial, with a challenge pertaining to the appropriate taxon sampling. However, with the genome data readily and publicly available, I believe that the effects of autotrophy versus simple body plan-changes could have been addressed, by e.g., including Lentibulariaceae (Genlisea: Leushkin et al. 2013. The miniature genome of a carnivorous

plant *Genlisea aurea* contains a low number of genes and short non-coding sequences. BMC Genomics 14: 476.; Utricularia: Ibarra-Laclette et al. 2013. Architecture and evolution of a minute plant genome. Nature 498: 94–98.). Although they complement their nutritional needs through carnivory, Lentibulariaceae are photosynthetic plants that underwent multiple (?) rounds of genome size reduction following polyploidization and that show extreme body plan modifications, incl. the loss of roots, among others. Expanding the analysis to include these plants (and several other lamiid for which genomic data is available) will thus help to substantiate several other claims by means of phylogenetic hypothesis testing.

The authors claim gene losses or losses/reductions of orthogroups, but according to their Materials & Methods section, no efforts were made to detect (highly) diverged fragments, for instance by explicitly searching missing genes that might have simply slipped automated annotation. This leaves several doubts as to whether that many genes were really lost. I am also somewhat surprised that the authors apparently did not care to further characterize the repetitive DNA fraction, which apparently makes up half the genome. Furthermore, positive selection was detected for a number of genes using a standard model, but this analysis fails to correct or account for effects of selection relaxation, especially the relaxation of purifying selection. We may assume the latter to be an important player in lineages where functional reconfigurations and ecological shifts have occurred in the past. Besides this, selecting *Cuscuta* as the only foreground branch a priori might have also comprised the analysis, because changes that might have set in earlier (e.g. ancestral to Convolvulaceae) will remain undetected and because the other taxa together are treated as background (see work by Kosakowski for details).

The authors would like to sell *Cuscuta* as “an excellent model to elucidate ... the evolution of plant parasitism” (line 36/37). The classification of *Cuscuta* as hemi- or holoparasite has been challenging in the past, and vivid debates are still ongoing. Applying a traditional definition of holoparasitism, which derives mainly from taxonomic viewpoints, *Cuscuta* spp., incl. *C. australis*, are rather “physiological holoparasites”. They often contain trace amounts of chlorophyll but cannot sustain themselves from their own photosynthesis, but there is at least one species that reportedly carries out measurable photosynthesis. Hence, the authors have sequenced a parasite, which may be at a transitional stage between hemiparasites and true holoparasitic plants (defined traditionally as lacking any photosynthetic pigments), but the authors ignore these biological/ecological facts in the interpretation of their data, and they provide no concluding physiological data to clarify the issue. From a plant genomicist’s viewpoint the choice of *C. australis* is understandable (small genome size), but not from a biologist’s. *Cuscuta* spp. have been reported to span an enormous range of genome sizes, with a weak correlation for genome size increases in nonphotosynthetic species (McNeal, J.R., et al. 2007. Systematics and plastid genome evolution of the cryptically photosynthetic parasitic plant genus *Cuscuta* (Convolvulaceae). BMC Biology 5: 55). Therefore, it remains to be established how representative the findings of this study really are, especially with regard to “lifestyle”-specific orthogroup reductions and “autotrophy” gene losses. From an evo-devo viewpoint, I find it disappointing that gene expression data was used only to help annotating the genome, but not for addressing the issue of the ontogenetic origin of haustoria in *Cuscuta*. It is still unclear from what tissue haustoria form, and experts still engage in the debate if these are modified roots or shoot structures. An in-depth analysis of expressed genes in haustoria thus might provide further insights into this topic, especially as these results could be nicely compared with existing work on haustorium development of root parasites in Orobanchaceae.

From the technical side of the paper, I would like to note that the assembly probably would have benefitted from hybrid approach that exploits all libraries and generated data (three different paired-end or mate pair Illumina data plus the PacBio long-read data), and I am wondering why this was omitted. The low amount of mapped transcript data of the study material compared to those from

another species also point toward some problems with these data type, and I am wondering if this might have compromised overall genome annotation. Regarding the loss of defense orthogroups, it must be kept in mind that the study material was grown for several generations under greenhouse conditions in the authors' labs after purchasing it from a commercial company. While five generation surely might not have contributed to shrinking defense mechanisms, it is unknown how long the plant was in cultivation and "under domestication" with this company. We know from many crop examples that severe resistance bottleneck are introduced during domestication and reduced/modified exposure to pathogens. The authors should validate R-gene reductions to rule out artifacts from cultivation and experimentation. It is also mentioned in the manuscript that R gene reduction is seen in *Ipomoea*, an autotrophic plant of the same family as *Cuscuta*. Thus, causation of R gene loss is unclear, and the alleged parasitism-effect may even be questioned in light of the findings in *Ipomoea* and the unknown time of dodder cultivation before seed purchase.

To conclude my major concerns: Although I find this work mostly technically sound concerning what has been done, many questions relating to parasitism or functional-genomics in *Cuscuta* remain unanswered and, unfortunately, even unaddressed. This is mainly due to the lack of a number of more sophisticated analyses or a wider evolutionary view to disentangle lifestyle effects from changes of the plant's body plan. However, also to the biology and ecology of *Cuscuta* was not considered in its details to allow for robust conclusions. Hence, several results and claims may not hold true when other methods are employed or when more taxa of the same or different lineages are added. My impression of this work is that a resource was generated but substantial biological or functional-evolutionary insights remain to be discovered (by others?). Therefore, many of the findings to me are purely descriptive, and various aspects of the manuscript are very speculative (e.g. evolution of "holoparasitic *Cuscuta* via leafy hemiparasitic *Cuscuta*; illustrated even as Fig. 4).

Apart from these major concerns, below is a set of minor points that may help to improve the manuscript:

L17: *Cuscuta* are at best a clade of parasites that transition from hemiparasitism to (physiological) holoparasitism, rendering many of the claims of this manuscript are speculative

L28-37: This description applies only to "haustorial parasites" and disregards the existence of mycoheterotrophs, which clearly also are parasitic but feed on fungi and develop no haustoria. Also, all points mentioned here fits all 12/13 lineages of haustorial parasites, of which at least three are better "qualified" as references for understanding the evolution and consequences of a parasitic lifestyle in plants.

L49: There was no doubt of the phylogenetic affinity of *Cuscuta*, and I'm wondering why the authors find that important to report, given that other analyses to validate their claims would have been more substantial to this study.

L68: "7Ref-species" to me sounds like lab jargon, and has never been introduced in this manuscript before.

L77 (and all other occurrences): species names incl. *Arabidopsis* should be italicized

L109: why were plastid genes not considered, too? The plastid chromosome appears quite large still, and readers, including this reviewer, surely would be interested to know how it fits into current models of reductive (plastome) evolution in parasitic plants (e.g., Wicke et al. 2016. Mechanistic model of evolutionary rate variation en route to a nonphotosynthetic lifestyle in plants. Proc. Natl. Acad. Sci. U.S.A. 113: 9045–9050 for a recent original paper, or Graham et al. 2017. Plastomes on the edge: the evolutionary breakdown of mycoheterotroph plastid genomes. New Phytol 214: 48–55 for a review, focusing on non-haustorial parasites though).

L128: R gene reduction thus might be a more general feature within Convolvulaceae, which must be considered for the discussion of these results; also see my major comment on R gene bottlenecks under long-term cultivation. Also, the number of genes really may be irrelevant as long as all critical and essential pathways are maintained.

L156: genomic changes might also be the results of extensive body plan reconfigurations (or vice versa), of which the twining habit could perhaps even be corrected for as Ipomea is a twiner, too.

L163: This form of (positive) feedback loop was described already by Wicke et al., 2016 (full reference above)

L258: Author contributions are missing for two or three of the eleven originally listed study authors.

L364: Is there any voucher of the material? The authors describe this genome sequence as a resource for future studies. How will plant germplasm be distributed to other researchers?

L377: How was imbibition of seeds and germination carried out?

L379: What method was used for RNA extractions?

L388/Assembly: Why no hybrid assembly using all paired-end/mate-pair libraries plus the long-read data? Assembly of transcript data: How were host-derived transcripts filtered (see e.g. LeBlanc et al. (2012) RNA trafficking in parasitic plant systems. Front. Plant Sci. 3: 203. for a review of molecule trafficking in parasitic plant/host systems)

L458: Why were hard or soft masked regions not characterized any further in light of gene losses and orthogroup reductions? (critical)

L598: Why were the 1887 lost orthogroups not specifically searched for with more targeted methods? (critical)

L683: Apparently no contaminant filtering was performed, and I wonder why no unigene-based orthogrouping was carried out to strengthen genome-based results, for which, however, have the genome was masked.

L689: I am somewhat surprised that transcripts from the same species yield considerably fewer hits than those of another species. What explains this discrepancy?

Dear Editor and Reviewers,

We would like to thank you for your suggestions and comments. We have carefully responded to all the concerns, comments, and suggestions. We believe that the quality and readability of this manuscript have been substantially improved after this revision.

Please see below our point-to-point response (in blue) to your comments. The changes in the manuscript are highlighted with yellow.

Thank you very much again for your suggestions, and we are looking forward to your decision on this revised manuscript.

Sincerely,

Jianqiang Wu

Reviewer #1 (Remarks to the Author):

Sun et al. report a high quality reference genome for the parasitic plant *Cuscuta australis*. Based on detailed comparative genomics analysis *C. australis* has experienced extreme gene loss presumably related to its parasitic lifestyle, including many genes related to photosynthesis, leaf, flower and root development among others. Many gene families are contracted including those containing R genes and P450 enzymes reflecting relaxed disease pressures. The manuscript is well written and exciting with major insights into genome evolution and parasitic plant biology. I have a few minor comments.

1. What is the nature of missing orthologous genes? If these genes have root or leaf specific expression, it's conceivable they would be absent from the trinity based transcriptome assembly and may be discarded by Pasa during annotation. The loss of flowering and circadian clock genes is particularly fascinating but also unusual, as these plants produce (relatively) normal flowers. The authors identified ~1,283 putative pseudogenes but it is unclear if these 'core' pathway genes have become pseudogenes or have been deleted altogether.

Response:

The reviewer concerned that the missing orthologous genes may be absent from transcriptome assembly and thus may be discarded during annotation by Pasa.

In addition to the transcriptome data, we also used de novo and homology-based methods in the annotation step which complement the potential problem of incomplete transcriptome information. Explicit validation including Genewise comparison, tblastn search as well as manual inspections (detail see Supplementary Note 3) were also performed to ensure that the missing orthologous genes were not a result of incomplete annotation.

Probably because we did not describe it very clearly, the flower development-related genes were not lost in *C. australis*, but some of the flowering time-related genes (Fig. 3b). Therefore, we speculate that *C. australis* probably senses the signals from their hosts (e.g., florigen) and synchronize its flowering time with that of the host. To avoid confusion, we stressed “flowering time genes” in the revised manuscript.

In our analysis, “lost genes” are those that can no longer be found in the genome, and pseudogenes were not included in the list of lost genes. The genes shown in Fig. 3b are all deleted genes, but not pseudogenes. Moreover, we inspected the list of pseudogenes (Supplementary Data 3a). Although GO analysis did not indicate enrichment of very interesting pathways, we did find five genes functioning in flowering time control and 15 photosynthesis-related genes. The most likely scenario is that the most of these lost genes in these core pathways, such as those controlling flowering time, were pseudogenized initially then gradually degenerated and merged into the background of the surrounding sequences and some of them were deleted from the genome.

2. The authors assembled a complete 116kb chloroplast genome as well as several other contigs. Other *Cuscuta* lineages have experienced wide scale gene loss (See Braukmann et al, J Exp Bot. 64(4); 977-989) and it would be interesting to report organelle gene loss in *C. australis*.

Response:

This is an important point. In the revised manuscript, we compared the chloroplast genome of *C. australis* with the genomes of four *Cuscuta* species (*C. obtusiflora* and *C.*

gronovii, which belong to the same subgenus *Grammica* with *C. australis*, and *C. exaltata* and *C. reflexa*, which belong to the subgenus *Monogynella*). Furthermore, the chloroplast genomes of root parasites *Orobanche cumana* and *Striga hermonthica* were also included (Supplementary Data 4). All these plastomes exhibit wide scale gene loss, and *ndh* genes were lost in all parasites. Please see lines 159-168 for details.

3. What program was used to align the Illumina data to the draft PacBio contigs prior to Pilon correction? How many indels/SNPs did Pilon correct? The estimated error rate is somewhat circular if Pilon was run based on the Illumina data aligned using BWA and GATK.

Response:

BWA was used previously to align the Illumina data to the draft PacBio assembly and the error rate was also estimated using BWA. As pointed out, this was a mistake. In the revised manuscript, we used bowtie2 and freebayes for estimation of the assembly error rate (see Assembly completeness, accuracy, and heterozygosity assessment in the Methods.), and 101,825 indels and 980 SNPs were corrected by Pilon (Supplementary Table 3). The accuracy was estimated to be 99.99% for the final assembly (Supplementary Table 3).

4. List the versions for all bioinformatics software (Canu, Quiver, Pilon, etc).

Response: The versions of all bioinformatics softwares have been added.

5. Lines 582-588. This is difficult to follow

Response: This part has been rewritten (“2. Identification of orthogroups from collinear fragment information” in Supplementary Note 3).

6. Line 666 drew to drawn

Response: We have corrected this mistake.

Reviewer #2 (Remarks to the Author):

1. The manuscript by Sun et al. describes the genome of the parasitic plant *Cuscuta australis*, and conducts a set of basic comparative genomic analyses. I realize that this

work presents an important resource for the field of plant genomics. However, I have several reservations about it to note, some of which of major concern to me. Several claims appear to be based on weak grounds. The authors claim that many of the genomic changes are due to parasitism, precisely to the loss of autotrophy and the “regressive evolution” associated with it. However, except for the ca. 60 photosynthesis gene orthologs that are indeed directly associated with autotrophy, many other changes such as substantial gene family expansions or reductions are not solely attributed to the autotrophic lifestyle but to an extensive change of a plant’s body plan. In consequence, the authors oversimplify their data and resulting statements/conclusions. Whether or not the evolution of these gene family truly associate with the lifestyle thus are in the open and it has not been formally tested, in part due to the authors’ study design. Disentangling lifestyle effects is everything but trivial, with a challenge pertaining to the appropriate taxon sampling. However, with the genome data readily and publicly available, I believe that the effects of autotrophy versus simple body plan-changes could have been addressed, by e.g., including Lentibulariaceae (Genlisea: Leushkin et al. 2013. The miniature genome of a carnivorous plant *Genlisea aurea* contains a low number of genes and short non-coding sequences. *BMC Genomics* 14: 476.; Utricularia: Ibarra-Laclette et al. 2013. Architecture and evolution of a minute plant genome. *Nature* 498: 94–98.). Although they complement their nutritional needs through carnivory, Lentibulariaceae are photosynthetic plants that underwent multiple (?) rounds of genome size reduction following polyploidization and that show extreme body plan modifications, incl. the loss of roots, among others. Expanding the analysis to include these plants (and several other lamiid for which genomic data is available) will thus help to substantiate several other claims by means of phylogenetic hypothesis testing.

Response:

This is an important point. We hypothesized that gaining parasitism was a step of evolution prior to the large body plan change in *Cuscuta*, as it is unlikely that *Cuscuta* ancestor heavily degenerated root and/or leaf before acquiring parasitism (this was also concluded in Fig. 4 in the previous manuscript, which has been removed in the revised version). Therefore, in the previous manuscript, we proposed that most of these gene loss events resulted from the change in lifestyle, namely from autotrophy to parasitism (the

large morphological changes were also a consequence of this lifestyle shift in *Cuscuta*). We agree with your comment that it is still too early to attribute body plan changes to parasitism. In the revised manuscript, we no longer stress that parasitism is the only reason for gene loss, but both parasitism and body plan changes.

Including *Utricularia gibba* in the analysis was an excellent idea. In the revised manuscript, we performed analyses on gene family contraction/expansion and gene loss. Interestingly, *U. gibba* genome also exhibits a large number of gene loss (and gene family contractions). We found that around 1/3 of the lost genes in *U. gibba* and *C. australis* are the same genes (lines 118). These gene loss events are likely to be resulted from their body plan changes, probably mainly degeneration of roots (lines 119). Based on transcriptomic data analysis, we found that the lost genes in *C. australis* are principally expressed in leaves and roots of tomato and Japanese morning glory plants (lines 129-136). Similarly, we found that the lost genes in *U. gibba* are also principally expressed in roots of tomato and Japanese morning glory, and so did the commonly lost genes of *U. gibba* and *C. australis*. However, tomato and Japanese morning glory are very remotely related with *U. gibba*, and there are no relatively complete *Mimus guttatus* (more related to *U. gibba* than the other species included in this work) transcriptomic data from different tissues available. In order to avoid controversy, in the revised manuscript we did not show the data of the principally expressed tissues of the lost genes in *U. gibba*.

In our analysis, we included seven autotrophic species (an outgroup *Arabidopsis thaliana*, and six lamiids plants – the Solanales *Ipomoea nil*, *Solanum tuberosum*, *Solanum lycopersicum*, and *Capsicum annuum*, the Gentianales *Coffea canephora*, and the Lamiales *Mimulus guttatus*). In terms of their relationship to *Cuscuta*, these species are from the same family (*Ipomoea nil*), the same order (two *Solanum* plants and *Capsicum*), and the same lamiid clade (*Mimus guttatus*), and finally the outgroup *Arabidopsis thaliana*. We believe that these seven species are enough for comparison with *Cuscuta*. Certainly, it would be very interesting to include more genomes from other parasitic plants, mycoheterotrophic, and carnivorous plants in the future.

2. The authors claim gene losses or losses/reductions of orthogroups, but according to their Materials & Methods section, no efforts were made to detect (highly) diverged fragments, for instance by explicitly searching missing genes that might have simply slipped automated annotation. This leaves several doubts as to whether that many genes were really lost.

Response:

Previously, we utilized three commonly used pipelines, homology-based, transcriptome-based, and *de novo* prediction, to annotate *C. australis* genome (lines 423); furthermore, pseudogenes were annotated with a homology-based pipeline from the masked genome. In the revised manuscript, we further improved the genome annotation: Genes in the orthogroups whose *C. australis* or *U. gibba* members are missing were used as the queries in tblastn and Genewise analysis on both the unmasked genome and the assembled unmappable *C. australis* transcripts. These two softwares do not require any previous mask and annotation, and thus can identify genes that were failed to be annotated in the previous pipelines. For details, please see Methods (Supplementary Note 3).

3. I am also somewhat surprised that the authors apparently did not care to further characterize the repetitive DNA fraction, which apparently makes up half the genome.

Response:

In the revised manuscript, we remasked the genome of *Ipomoea nil* (as it is the most closely related) with the same standards used for *C. australis* genome and compared their repetitive regions. The data are shown in Supplementary Table 4.

The repeat contents were found to be 58 and 64.5% for the genomes of *C. australis* and *I. nil*, respectively, and different types of repeats, including LTR, DNA transposons, and simple repeats, have similar contents in these two plant species (Supplementary Table 4). Since Gypsy and Copia are two important subfamilies of LTRs, we specifically determined their copy numbers and sequence divergence in *C. australis* and *I. nil*. Details are described in lines 416-422 and Supplementary Figure 1.

4. Furthermore, positive selection was detected for a number of genes using a standard model, but this analysis fails to correct or account for effects of selection relaxation, especially the relaxation of purifying selection. We may assume the latter to be an important player in lineages where functional reconfigurations and ecological shifts have occurred in the past. Besides this, selecting *Cuscuta* as the only foreground branch a priori might have also comprised the analysis, because changes that might have set in earlier (e.g. ancestral to Convolvulaceae) will remain undetected and because the other taxa together are treated as background (see work by Kosakovski for details).

Response:

We agree with the comments on positive selection and relaxed selection. In the revised manuscript, we redid the analysis on positive selection using the methods from Kosakovski to obtain selection patterns for all branches in all orthogroup trees. For a given gene, if the positively selection was not specific to *Cuscuta*, namely, positive selection could also be found in the ancestral branches, this gene was no longer considered to be positively selected. Genes that showed relaxed purifying selection were also removed from the positively selected genes.

3. The authors would like to sell *Cuscuta* as “an excellent model to elucidate ... the evolution of plant parasitism” (line 36/37). The classification of *Cuscuta* as hemi- or holoparasite has been challenging in the past, and vivid debates are still ongoing. Applying a traditional definition of holoparasitism, which derives mainly from taxonomic viewpoints, *Cuscuta* spp., incl. *C. australis*, are rather “physiological holoparasites”. They often contain trace amounts of chlorophyll but cannot sustain themselves from their own photosynthesis, but there is at least one species that reportedly carries out measurable photosynthesis. Hence, the authors have sequenced a parasite, which may be at a transitional stage between hemiparasites and true holoparasitic plants (defined traditionally as lacking any photosynthetic pigments), but the authors ignore these biological/ecological facts in the interpretation of their data, and they provide no concluding physiological data to clarify the issue.

Response:

In the revised manuscript, we provided the data of photosynthesis in *C. australis* (Supplementary Note 8 and Supplementary Fig. 7). Although we used the GFS-3000 system (Heinz-Walz Instruments, Effeltrich, Germany), which is one of the most sensitive instruments, the photosynthesis activity of *C. australis* was below the sensitivity limit. Probably a very specially designed instrument is needed to detect its photosynthesis activity. However, photosynthesis II fluorescence measurement showed that *C. australis* has photosynthesis activity, although at a very low level (Supplementary Fig. 7). Since photosynthesis measurement is not central to this work, we did not pursue any further measurement of *C. australis* photosynthesis.

We agree that it is debatable whether *Cuscuta* species are hemi- or holoparasites. However, *Cuscuta* is still a nice system to study the evolution of parasitism and host-parasite interactions. “Excellent” has been removed and this part has been revised (lines 45-46).

4. From a plant genomicist’s viewpoint the choice of *C. australis* is understandable (small genome size), but not from a biologist’s. *Cuscuta* spp. have been reported to span an enormous range of genome sizes, with a weak correlation for genome size increases in nonphotosynthetic species (McNeal, J.R., et al. 2007. Systematics and plastid genome evolution of the cryptically photosynthetic parasitic plant genus *Cuscuta* (Convolvulaceae). *BMC Biology* 5: 55). Therefore, it remains to be established how representative the findings of this study really are, especially with regard to “lifestyle”-specific orthogroup reductions and “autotrophy” gene losses.

Response:

The genome sizes of *Cuscuta* species are very different, but so do the plants of various other genera. It is very common that plants from the same genus have rather different genome sizes, mainly due genome duplications. For genome analysis, technically it is preferred to start with the species that have relatively small genome sizes, which usually increases the chances of obtaining high quality genome sequences.

We chose *C. australis* for genome sequencing. The most important reason is that it has a small genome, making genome sequencing and assembly relatively simple. We would like to sequence a few other species of *Cuscuta* in the future and comparative

genomics will be done to further study gene loss and potential parasitism-related genes. Certainly, combining genome information from other parasitic plants in the comparative analysis would also be very important in the future.

5. From an evo-devo viewpoint, I find it disappointing that gene expression data was used only to help annotating the genome, but not for addressing the issue of the ontogenetic origin of haustoria in *Cuscuta*. It is still unclear from what tissue haustoria form, and experts still engage in the debate if these are modified roots or shoot structures. An in-depth analysis of expressed genes in haustoria thus might provide further insights into this topic, especially as these results could be nicely compared with existing work on haustorium development of root parasites in Orobanchaceae.

Response:

This is an important point. In the revised manuscript, we analyzed the *Cuscuta* transcriptomic data. Firstly, we found that the genes of the orthogroups, which have lost their *Cuscuta* orthologs (gene loss), are principally expressed in the root and leaf tissues of *Solanum lycopersicum* and *Ipomoea nil*. This result is consistent with the body plan of *Cuscuta*, namely, root- and leafless. Please see lines 129-136 for details.

Next, we also analyzed the principally expressed genes (PEGs) in *Cuscuta* prehaustoria/haustoria. These PEGs' orthogroups were subsequently identified, and were used to identify the orthogroups' principally expressed tissues in *S. lycopersicum* and *I. nil*. We found that the principally expressed tissues were roots (Supplementary Fig. 9). These data imply that the evolution of *Cuscuta* haustorium may be related to expression changes of genes involved in root development, and there are certain similarities between the evolution of haustoria in *Cuscuta* and Orobanchaceae root parasites. Please see lines 219-225 for details.

6. From the technical side of the paper, I would like to note that the assembly probably would have benefitted from hybrid approach that exploits all libraries and generated data (three different paired-end or mate pair Illumina data plus the PacBio long-read data), and I am wondering why this was omitted. The low amount of mapped transcript data of the study material compared to those from another species also point toward some problems

with these data type, and I am wondering if this might have compromised overall genome annotation.

Response:

Previously, we did not use the mate-pair Illumina data to assist scaffolding the PacoBio contigs. In the revised manuscript, the mate-pair libraries were used to build scaffolds (see Methods). Furthermore, PacBio reads were also aligned against the assembled scaffolds, but no linking information was found.

In the Methods of the previous version, we wrote that “Reads from *Cuscuta pentagona* transcriptome were aligned against *Cuscuta australis* genome using HISAT2 with the relaxed parameters “-mp 3,1”, achieving an average ratio of the mapped reads of 73.42%. However, nearly 91% of *Cuscuta pentagona* assembled transcripts could be aligned with an average identity of 96% (Supplementary Table 9), suggesting high identity of coding regions between the two species”. All these mapping ratios were actually from *C. pentagona*. We made a mistake by not including *C. australis* transcript mapping ratio data. In the revised manuscript, the *C. australis* RNA-seq reads, in which the reads from hosts were removed, were mapped to the genome and the ratio reached 96%. We also filtered out the host-derived reads from *C. pentagona* RNA-seq reads and after mapping them to *C. australis* genome, the mapping ratio was found to be 75%. Please see Supplementary Note 7 for details.

7. Regarding the loss of defense orthogroups, it must be kept in mind that the study material was grown for several generations under greenhouse conditions in the authors’ labs after purchasing it from a commercial company. While five generation surely might not have contributed to shrinking defense mechanisms, it is unknown how long the plant was in cultivation and “under domestication” with this company. We know from many crop examples that severe resistance bottleneck are introduced during domestication and reduced/modified exposure to pathogens. The authors should validate R-gene reductions to rule out artifacts from cultivation and experimentation. It is also mentioned in the manuscript that R gene reduction is seen in *Ipomoea*, an autotrophic plant of the same family as *Cuscuta*. Thus, causation of R gene loss is unclear, and the alleged parasitism-

effect may even be questioned in light of the findings in *Ipomoea* and the unknown time of dodder cultivation before seed purchase.

Response:

In China, *Cuscuta* seeds can easily be purchased from Chinese traditional medicinal stores or internet. However, *Cuscuta* is not considered to be an important medicine (it is rarely used). As far as we know, there is no large demand for *Cuscuta* seeds, and they are very rarely cultivated. Most of the seeds are collected from naturally grown *Cuscuta* plants. Even occasionally some seeds may be collected from cultivated *Cuscuta*, the *Cuscuta* plants did not experience any artificial selection on any traits (*Cuscuta* is even very hard to cross), including disease resistance, let alone selecting for decreased R genes.

Our *C. australis* was initially purchased from a Chinese traditional medicinal store and was inbred for five generations, as inbreeding for a few generations is a very commonly used means to improve homozygosity of wild species, in case they do have relatively high levels of heterozygosity. We always randomly chose a single plant to harvest seeds for the next generation, and we did not select for any traits that could possibly decrease the number of R genes. Usually plants have many R genes, which are distributed in almost all chromosomes. It is very hard to imagine that many of them could be artificially removed by inbreeding just for a few generations. Actually, *Ipomoea nil*, *Solanum tuberosum*, *Solanum lycopersicum*, *Capsicum annum*, and *Coffea canephora*, whose R genes were analyzed in this manuscript, have a long history of been cultivated, but they still maintain a large number of R genes.

As roots and leaves are the most common infection sites, after *Cuscuta* became root- and leafless, most pathogens can no longer infect *Cuscuta*, and this had led to relaxed selection pressure on these R genes (lines 190-195). We believe that this is the most likely reason for the remarkable gene loss in R genes. Likely for the same reason, *EDS1*, *EDS5*, *FMO1*, *SAG101*, and *PAD4*, which are important genes for plant resistance to diseases, and *ALDI*, which is critical for systemic acquired resistance, are also missing in *C. australis* genome (lines 203-206).

We agree that there is a possibility that R gene reduction took place within Convolvulaceae (this point has been added to the revised manuscript. Please see lines 190-193). However, the R genes in *C. australis* is still substantially smaller than those in *Ipomoea nil*.

8. To conclude my major concerns: Although I find this work mostly technically sound concerning what has been done, many questions relating to parasitism or functional-genomics in *Cuscuta* remain unanswered and, unfortunately, even unaddressed. This is mainly due to the lack of a number of more sophisticated analyses or a wider evolutionary view to disentangle lifestyle effects from changes of the plant's body plan. However, also to the biology and ecology of *Cuscuta* was not considered in its details to allow for robust conclusions. Hence, several results and claims may not hold true when other methods are employed or when more taxa of the same or different lineages are added. My impression of this work is that a resource was generated but substantial biological or functional-evolutionary insights remain to be discovered (by others?). Therefore, many of the findings to me are purely descriptive, and various aspects of the manuscript are very speculative (e.g. evolution of “holoparasitic *Cuscuta* via leafy hemiparasitic *Cuscuta*; illustrated even as Fig. 4).

Response:

In the revised manuscript, we included the genome of *Utricularia gibba* for the analysis of gene family contraction/expansion and gene loss. More in-depth transcriptomic data analysis was also done. We believe that this new version of the manuscript provides more insight into the evolution of *Cuscuta*. A more stringent pipeline was also used for gene loss identification, which minimized the risk of false positives (see Supplementary Note 3 for details).

We agree that whether the pattern of *Cuscuta* genome evolution (such as gene loss) holds true for other parasites is unclear, but we would like not to include more genomes in our analysis. We are looking forward to publication of genomes from other parasitic species. Comparative genomics will tell us whether there are any similarities between the genome evolution of *Cuscuta* and other parasitic plants.

Fig. 4 in the previous version of the manuscript has been removed.

Minor concerns:

L17: *Cuscuta* are at best a clade of parasites that transition from hemiparasitism to (physiological) holoparasitism, rendering many of the claims of this manuscript are speculative

Response: This point has been introduced in the revised manuscript (lines 40-43).

L28-37: This description applies only to “haustorial parasites” and disregards the existence of mycoheterotrophs, which clearly also are parasitic but feed on fungi and develop no haustoria. Also, all points mentioned here fits all 12/13 lineages of haustorial parasites, of which at least three are better “qualified” as references for understanding the evolution and consequences of a parasitic lifestyle in plants.

Response:

“Haustorial parasites” have been added to this part (lines 33-36).

We agree that there are many more species probably qualified better than *C. australis* for understanding the evolution of parasitism. However, clearly it is out of the scope of this manuscript to include other parasitic plant species, as *C. australis* genome is the first parasitic plant genome and genome sequences from any other parasitic plants are not available yet. We are looking forward to seeing more interesting genomes of parasitic plants to be published.

L49: There was no doubt of the phylogenetic affinity of *Cuscuta*, and I’m wondering why the authors find that important to report, given that other analyses to validate their claims would have been more substantial to this study.

Response:

We agree with the reviewer that the phylogeny of *Cuscuta* and other species is clear. One reason is that presenting a phylogeny constructed with whole genome one-to-one orthogroups is quite a routine in genome papers. If we do not present this tree, other reviewers and readers will question why this is not shown. Another important reason is that with the whole genome data available, we could reconstruct a species tree with more accurate branch lengths, in which we found that *C. australis* experienced rapid evolution.

The lengths of the branches of the phylogenetic tree inferred are also required for the analysis of gene family contraction and expansion.

L68: "7Ref-species" to me sounds like lab jargon, and has never been introduced in this manuscript before.

Response:

The definition of "7Ref-species" has been described on its first appearance in line 79. We cannot find a way to avoid using this alias. If we do not use this alias, but list all the species names, many sentences will become very long and not succinct to read. Therefore, we would like to retain "7Ref-species".

L77 (and all other occurrences): species names incl. *Arabidopsis* should be italicized

Response: We have italicized species names.

L109: why were plastid genes not considered, too? The plastid chromosome appears quite large still, and readers, including this reviewer, surely would be interested to know how it fits into current models of reductive (plastome) evolution in parasitic plants (e.g., Wicke et al. 2016. Mechanistic model of evolutionary rate variation en route to a nonphotosynthetic lifestyle in plants. *Proc. Natl. Acad. Sci. U.S.A.* 113: 9045–9050 for a recent original paper, or Graham et al. 2017. Plastomes on the edge: the evolutionary breakdown of mycoheterotroph plastid genomes. *New Phytol* 214: 48–55 for a review, focusing on non-haustorial parasites though).

Response:

This is an important point. In the revised manuscript, we compared the chloroplast genome of *C. australis* with the genomes of four *Cuscuta* species (*C. obtusiflora* and *C. gronovii*, which belong to the same subgenus *Grammica* with *C. australis*, and *C. exaltata* and *C. reflexa*, which belong to the subgenus *Monogynella*). Furthermore, the chloroplast genomes of root parasites *Orobanche cumana* and *Striga hermonthica* were also included (Supplementary Data 4). All these plastomes exhibit wide scale gene loss, and *ndh* genes were lost in all parasites. Please see lines 159-168 for details.

L128: R gene reduction thus might be a more general feature within Convolvulaceae, which must be considered for the discussion of these results; also see my major comment

on R gene bottlenecks under long-term cultivation. Also, the number of genes really may be irrelevant as long as all critical and essential pathways are maintained.

Response: Please see our response to Comment 7.

L156: genomic changes might also be the results of extensive body plan reconfigurations (or vice versa), of which the twining habit could perhaps even be corrected for as *Ipomea* is a twiner, too.

Response:

We agree with this point. Please see our response to Comment 1. In this manuscript, we do not intend to analyze the twining habit of *Cuscuta* or *Ipomea*, but focusing on the genomic comparisons between *Cuscuta* and autotrophic plants (*Ipomea* is only considered as a closely related autotrophic species from the same family).

L163: This form of (positive) feedback loop was described already by Wicke et al., 2016 (full reference above)

Response: We have cited Wicke's elegant work in the manuscript.

L258: Author contributions are missing for two or three of the eleven originally listed study authors.

Response: This has been corrected in the revised manuscript.

L364: Is there any voucher of the material? The authors describe this genome sequence as a resource for future studies. How will plant germplasm be distributed to other researchers?

Response:

Voucher specimens of *C. australis* can be accessed at the Herbarium of Kunming Institute of Botany, Chinese Academy of Sciences (accessions Nos. WJQ-001-1 and WJQ-001-3). *C. australis* seeds can be distributed by Dr. Jianqiang Wu (Kunming Institute of Botany, Chinese Academy of Sciences, email: wujianqiang@mail.kib.ac.cn, up request. These lines of information have been added to the manuscript (lines 321-325).

L377: How was imbibition of seeds and germination carried out?

Response: The method of germination has been added to the manuscript (See Methods DNA sample preparation and sequencing).

L379: What method was used for RNA extractions?

Response:

RNA was extracted using the TRI Reagent. This has been added to the Methods (line 337).

L388/Assembly: Why no hybrid assembly using all paired-end/mate-pair libraries plus the long-read data? Assembly of transcript data: How were host-derived transcripts filtered (see e.g. LeBlanc et al. (2012) RNA trafficking in parasitic plant systems. Front. Plant Sci. 3: 203. for a review of molecule trafficking in parasitic plant/host systems)

Response:

In the revised manuscript, we used mate-pair libraries to build hybrid assembly, and the PacBio reads were also aligned against the assembled scaffolds but no linking information was found. For detail see Methods “De novo assembly”.

We also used the genome information of the hosts to filter out the likely transported mRNAs from the hosts (for details see Supplementary Note 7). Thus, the risk that the transcriptomic data contain “contamination” from hosts is minimized.

L458: Why were hard or soft masked regions not characterized any further in light of gene losses and orthogroup reductions? (critical)

Response:

In the revised manuscript, we remasked the genome of *Ipomoea nil* (as it is the most closely related) with the same standards used for *C. australis* genome and compared their repetitive regions. The data are shown in Supplementary Table 4.

The repeat contents were found to be 58 and 64.5% for the genomes of *C. australis* and *I. nil*, respectively, and different types of repeats, including LTR, DNA transposons, and simple repeats, have similar contents in these two plant species (Supplementary Table 4). Since Gypsy and Copia are two important subfamilies of LTRs,

we specifically determined their copy numbers and sequence divergence in *C. australis* and *I. nil*. Details are described in lines 57-69 and Supplementary Fig. 1.

L598: Why were the 1887 lost orthogroups not specifically searched for with more targeted methods? (critical)

Response:

Previously, we utilized three commonly used pipelines, homology-based, transcriptome-based, and *de novo* prediction, to annotate *C. australis* genome (lines 423); furthermore, pseudogenes were annotated with a homology-based pipeline from the masked genome. In the revised manuscript, we further improved the genome annotation: Genes in the orthogroups whose *C. australis* and *U. gibba* members are missing were used as the queries in tblastn and Genewise analysis on both the unmasked genome and the assembled unmappable *C. australis* transcripts. These two softwares do not require any previous mask and annotation, and thus can identify genes that were failed to annotate in the previous pipelines. For details, please see Methods (Supplementary Note 3).

L683: Apparently no contaminant filtering was performed, and I wonder why no unigene-based orthogrouping was carried out to strengthen genome-based results, for which, however, have the genome was masked.

Response:

We removed the host-derived contaminant transcripts in the RNA-seq data (Supplementary Note 7). Since transcript data are likely not as completely as genome data, for the analysis of gene loss, we only used the genome data, but not transcriptome data, to construct orthogroups. The risk that the genes that were not annotated in the genome but can be found in RNA-seq data were minimized by using our new gene loss analysis pipeline (see Supplementary Note 4).

L689: I am somewhat surprised that transcripts from the same species yield considerably fewer hits than those of another species. What explains this discrepancy?

Response:

The different mapping ratios were not actually not from *C. australis* and *C. pentagona* RNA-seq data, but from the same *C. pentagona* dataset run by two mapping softwares (TopHat for RNA-Seq data and NUCmer for assembled transcripts). We have rewritten this section and provided the mapping ratio table (Supplementary Table 8) of all samples including both *C. australis* and *C. pentagona* with the same mapping method.

REVIEWERS' COMMENTS:

Reviewer #1 (Remarks to the Author):

The authors have addressed my previous concerns and I feel this manuscript is suitable for publication.

Reviewer #2 (Remarks to the Author):

In their revised manuscript, Sun et al. provide a restructured work that benefits notably from additional analyses and a major re-write. Below are a several more comments that I believe would help to further improve the manuscript:

Comments regarding specific statements/claims:

L25: plain numbers are meaningless if no context is given. for example, 19,800 might seem not like many, but if the number of conserved genes in all sequenced plants (ca. 100) is only 22,000, the difference is marginal, given that a BUSCO analysis usually finds 5-10 % missing (which should be slightly higher in *Cuscuta*)

L57: Considering only the nuclear sequences or nuclear plus organelle data?

L72; L368: a BUSCO analysis should be performed to see how many of the 1440 land plant genes are found in *C. australis* and the 7Ref set. This will give readers an understanding of how meaningful the plain number of annotated genes is. Although a CEGMA analysis regarding the completeness in reference to eukaryotes has been performed, a BUSCO analysis in the context of herein reported data is more relevant, despite the fact that no details of the CEGMA analysis are provided in the main text. Paragraph starting L108: the definition that "conserved" is "present in at least 6 out of the 7 reference taxa" > this, together with BUSCO will provide the reader a better understanding of the magnitude of gene losses in "conserved" genes

Paragraph starting L137: Does this all include the losses in *U. gibba*, or losses shared between *C. australis* and *U. gibba*?

L167: It's more generally established that plastid *ndh* genes are the first to be lost along the transition to parasitism, so it's not only restricted to Orobanchaceae"

L181: There is absolutely no evidence that *Cuscuta* synchronizes its flowering with the flowering time of its host(s) – eavesdropping on host FT seems a little farfetched for my taste unless you show that *Cuscuta* has lost its own FTs.

L188: do you mean "..., reduced exposure to pathogens and insects relaxed selective constraints on R genes, resulting in their eventual loss"?

L481: Unless the authors guarantee the uninterrupted service and permanent maintenance of this custom website, I would like to strongly encourage the authors to deposit their files (assembly and raw read data for genome and transcriptome analysis, gene models, but also gene alignments used for selection analyses, phylogenetic trees, pseudogene annotations, etc., in public repositories that are long-term resources. As of performing this review, neither the NCBI data link ("Bioproject does not exist") worked, nor the authors' website.

Supporting figure S1: Some Gypsy and Copia copies have super-long branches. It might be worth checking if these are more closely related to other plants (potential hosts).

Some language suggestions:

L22: "accelerated rates of molecular evolution" instead of "accelerated evolution"

L27: "the massive changes of its body plan"

L30: "haustorium formation requires mostly genes normally involved in root development"

L39: "*Cuscuta* sp. exhibit massive changes of their body plans, ..."

L62: drop "does the" and "genome"

L63: "LTR retrotransposons" instead of "LTRs"

L110: "considerable" instead of "strong"

L162: "all of these"

L163: "lack ndh genes ..."

L226: " either "performed a HYPHY analysis" or "performed HYPHY analyses"

L240: new paragraph before "Gene" – gives readers a logical break; please also check the construct "ABC transporter B family member 15-like protein" if it can be simplified.

L249: there is no such thing as "positive evolution", so better change the entire phrase to "experienced relaxed purifying and/or positive selection..."

L262: "derived" implies that it is modified root and floral tissue (which it is not ontogenetically), so better change the phrase to "Orobanchaceae parasites recruit genes normally involved in the development of root and floral tissues"

L267: "... that the *C. australis* genome..."

L294: full stop after "new genes" and begin a new sentence "It is likely ..."

L296: "have experienced" or "have undergone"

L305: drop "genome" after "*U.gibba*"

References not checked

L675: "upon request"

Dear Reviewers and Editor,

I would like to thank you for your comments and suggestions on our previous version of the manuscript, “Large-scale gene losses underlie the genome evolution of parasitic plant *Cuscuta australis*”. These comments and suggestions have greatly helped us to improve the quality of this work.

Please find our responses to the comments below in blue. **Please note that all the line numbers mentioned below are the numbers in the version with track-changes, but not in the clean version.**

Reviewer #1 (Remarks to the Author):

The authors have addressed my previous concerns and I feel this manuscript is suitable for publication.

Reviewer #2 (Remarks to the Author):

In their revised manuscript, Sun et al. provide a restructured work that benefits notably from additional analyses and a major re-write. Below are several more comments that I believe would help to further improve the manuscript:

Comments regarding specific statements/claims:

- 1) L25: plain numbers are meaningless if no context is given. for example, 19,800 might seem not like many, but if the number of conserved genes in all sequenced plants (ca. 100) is only 22,000, the difference is marginal, given that a BUSCO analysis usually finds 5-10 % missing (which should be slightly higher in *Cuscuta*)

Response: In the revised manuscript, “only” has been removed and this sentence has been modified to be “*C. australis* genome harbors 19671 protein-coding genes, and importantly, 11.7% of the conserved orthologs in autotrophic plants are lost in *C. australis*”. Please note that the gene number in the previous version of this manuscript was 19805, which is wrong. We have changed it to 19617, as shown in page 4 line 10. 19805 was the gene number that we obtained from the

very early stage of the genome analysis, and we missed changing it during writing this manuscript. We apologize for this mistake.

- 2) L57: Considering only the nuclear sequences or nuclear plus organelle data?

Response: These are nuclear genome data, and “nuclear” has been added in front of “genome” (page 3 line 22).

- 3) L72; L368: a BUSCO analysis should be performed to see how many of the 1440 land plant genes are found in *C. australis* and the 7Ref set. This will give readers an understanding of how meaningful the plain number of annotated genes is. Although a CEGMA analysis regarding the completeness in reference to eukaryotes has been performed, a BUSCO analysis in the context of herein reported data is more relevant, despite the fact that no details of the CEGMA analysis are provided in the main text.

Response: This is an important suggestion. We performed BUSCO analysis on *C. australis*, *U. gibba*, and 7Ref-Species. As expected, the missing BUSCOs in *C. australis* and *U. gibba* (16.30% and 13.70%, respectively) are more than those in the 7Ref-Species (1.40% to 8.50%). This has been added to the manuscript (page 5 line 22 to page 6 line 4). Since CEGMA has been discontinued (<http://korflab.ucdavis.edu/Datasets/cegma/>) and the results from BUSCO analysis are presented, we removed the description of CEGMA in the Methods.

- 4) Paragraph starting L108: the definition that “conserved” is “present in at least 6 out of the 7 reference taxa” > this, together with BUSCO will provide the reader a better understanding of the magnitude of gene losses in “conserved” genes

Response: Yes, BUSCO analysis has been added to the manuscript (see above please).

- 5) Paragraph starting L137: Does this all include the losses in *U. gibba*, or losses shared between *C. australis* and *U. gibba*?

Response: As pointed out by the Editor, the section titles are not allowed to have more than 60 characters, including spaces. Thus, we could not add more specific information to the title. However, in the first paragraph, the readers can clearly see that here we are talking about *C. australis* only: “Next, the *C. australis* genome was specifically searched for genes that mediate leaf and root development, and it was found that a number of important genes involved in leaf and/or root development are absent....”.

The same for the next sections, “Loss of photosynthesis genes”, “Loss of genes controlling flowering time”, and “Loss of defense genes”.

Due to the limitation of characters in the section titles, we had to split the formerly “Loss of genes important for leaf and root development and nutrient uptake” section into two consecutive sections, which are now “Loss of genes for leaf and root development” and “Loss of genes for nutrient uptake”.

- 6) L167: It’s more generally established that plastid *ndh* genes are the first to be lost along the transition to parasitism, so it’s not only restricted to Orobanchaceae”

Response: This sentence has been modified: “These data concur with the generally accepted notion that *ndh* genes are the first to be lost in the initial stage in the evolution of parasitism, apparently due to relaxed selective constraint”.

- 7) L181: There is absolutely no evidence that *Cuscuta* synchronizes its flowering with the flowering time of its host(s) – eavesdropping on host FT seems a little farfetched for my taste unless you show that *Cuscuta* has lost its own FTs.

Response: This sentence has been deleted.

- 8) L188: do you mean “..., reduced exposure to pathogens and insects relaxed selective constraints on R genes, resulting in their eventual loss”?

Response: Yes, this sentence has been modified accordingly (page 9 lines 6-8).

- 9) L481: Unless the authors guarantee the uninterrupted service and permanent maintenance of this custom website, I would like to strongly encourage the

authors to deposit their files (assembly and raw read data for genome and transcriptome analysis, gene models, but also gene alignments used for selection analyses, phylogenetic trees, pseudogene annotations, etc., in public repositories that are long-term resources. As of performing this review, neither the NCBI data link (“Bioproject does not exist”) worked, nor the authors' website.

Response: The genome assembly, gene models, and sequence reads have been deposited at the NCBI under the BioProject PRJNA394036 [<https://www.ncbi.nlm.nih.gov/bioproject/PRJNA394036>], and the former two can also be accessed at <http://www.dodderbase.org>.

We did not choose to release the data immediately after depositing them at the NCBI, but we selected the option that they will be released after this manuscript is published officially (will be done by the NCBI staff). The website <http://www.dodderbase.org> is maintained by the IT center at the Kunming Institute of Botany, Chinese Academy of Sciences, which is a permanent website as well. This website will be accessible within 48 h after we receive the email notifying the acceptance of this manuscript (very likely the release of these data from this website will be earlier than from the NCBI database).

As suggested, the phylogenetic analysis data for detection of gene losses, gene alignments used for selection analysis, and pseudogene annotations can be accessed at <https://doi.org/10.6084/m9.figshare.6072131>.

Please see “Data Availability” for details.

- 10) Supporting figure S1: Some Gypsy and Copia copies have super-long branches. It might be worth checking if these are more closely related to other plants (potential hosts).

Response: This is a great suggestion. We have been performing analyses on the possible horizontally transferred genes in *C. australis* on a whole-genome level. Transposons are one of the important classes of candidate horizontally transferred genes, including Gypsy and Copia. Since the analyses are complex and the results

are too much to present in this manuscript (also out of the scope of this work), they will be presented in a separate manuscript, which we suppose could be submitted within a few months.

Some language suggestions:

L22: “accelerated rates of molecular evolution” instead of “accelerated evolution”

L27: “the massive changes of its body plan”

L30: “haustorium formation requires mostly genes normally involved in root development”

L39: “*Cuscuta* sp. exhibit massive changes of their body plans, ...”

L62: drop “does the” and “genome”

L63: “LTR retrotransposons” instead of “LTRs”

L110: “considerable” instead of “strong”

L162: “all of these”

L163: “lack *ndh* genes ...”

L226: ” either “performed a HYPHY analysis” or “performed HYPHY analyses”

L240: new paragraph before “Gene” – gives readers a logical break; please also check the construct “ABC transporter B family member 15-like protein” if it can be simplified.

L249: there is no such thing as “positive evolution”, so better change the entire phrase to “experienced relaxed purifying and/or positive selection...”

L262: “derived” implies that it is modified root and floral tissue (which it is not ontogenetically), so better change the phrase to “Orobanchaceae parasites recruit genes normally involved in the development of root and floral tissues”

L267: ”... that the *C. australis* genome...”

L294: full stop after “new genes” and begin a new sentence “It is likely ...”

L296: “have experienced” or “have undergone”

L305: drop “genome” after “*U.gibba*”

References not checked

L675: “upon request”

Response: Thanks for these suggestions. These language suggestions have all be incorporated in the manuscript.